# SchulzNN: A Neural Network-Based Matrix Inversion Solver Inspired by Schulz Iteration

## Abstract

While neural networks have emerged as powerful tools for solving optimization problems, demonstrating performance comparable to or surpassing traditional solvers, their application to fundamental numerical optimization problems remains underexplored. Specifically, limited progress has been made in developing neural network-Based approaches for matrix inversion – a cornerstone problem in unconstrained numerical optimization. This work presents SchulzNN, the first neural network-Based solver for matrix inversion inspired by the Schulz iterative method. Our architecture innovatively simulates traditional iteration processes through parametric learning, preserving theoretical convergence guarantees while enhancing computational flexibility. We rigorously evaluate both single-layer SchulzNN and deep variants on (1) diverse matrix families with practical significance, and (2) matrices beyond the convergence conditions of classical Schulz iteration. Notably, we establish a systematic framework for analyzing model adaptation through fine-tuning strategies. The accuracy and efficiency of the proposed SchulzNN are demonstrated by numerical examples for matrix inversion.

## 1 Introduction

In recent years, "Learn to Optimize" (L2O) Tang & Yao (2024), as an emerging optimization method, has made significant progress in both research and practical applications. One branch of L2O emphasizes the integration of machine learning with traditional optimization techniques Xin et al. (2021), proposing a data-driven framework for the design of optimization algorithms. This approach enables the automation of algorithm design, reducing the need for labor-intensive iterations, and is particularly suitable for scenarios where similar optimization problems need to be solved repeatedly on specific data distributions. Another branch focuses on an end-to-end learning approach Sun & Yang (2023) , where neural networks directly predict the solution to optimization problems, achieving success in many combinatorial optimization problems. However, when it comes to the most fundamental unconstrained optimization problem — computing the matrix inverse — neural network-based methods have seen limited success due to the inherent difficulty of solving this problem.

Finding the inverse of a matrix is a popular and interesting topic with a wide range of applications in real life, such as the calculation of the current of a wireless power transfer system Alberto & Brox (2020), problems of elastostatics and of vibration theory Wittenburg (1998), electric net design Guo et al. (2012), etc. The Moore-Penrose generalized inverse is a conceptual extension of the inverse of a matrix, and solving it is another important topic. For instance, optimizing the generalized inverse matrix could achieve hyperspectral image super-resolution Lin et al. (2024). The technics of generalized matrix can be applied to robot control system as well Zhang & Uhlmann (2018).Another notable application scenario lies in the extraction of the diagonal of a matrix inverse, particularly in domains such as electronic structure theory and electrostatic modeling, where researchers have devoted significant efforts to investigating efficient methods for computing the diagonal elements of a matrix inverse Tu et al. (2022; 2025).

In this paper, we propose the Schulz Neural Network (SchulzNN), a deep learning architecture designed to compute matrix inverses. By integrating Schulz iteration with neural networks, SchulzNN provides an innovative framework to address the longstanding challenge of matrix inversion. The contributions of this work are summarized as follows:

- Designed within the framework of Schulz iteration, SchulzNN autonomously approximates matrix inverses through a specialized training process. Its architecture naturally embeds both single-layer and multi-layer structures, leveraging the inherently iterative nature of the Schulz method. SchulzNN computes inverses for diverse matrix classes, including cases where traditional Schulz iteration exhibits significant limitations.
- We employ interpolation decomposition and butterfly factorization Liu & Yang (2020) to reduce the computational complexity of matrix-vector multiplication. By integrating these factorizations into the network architecture and leveraging the fact that matrix $A$ remains fixed in the neural network, we achieve an overall training complexity of $O(N^2 \log N)$ per epoch, representing a significant improvement over the conventional $O(N^3)$ complexity for matrix inversion.
- Given a matrix, its inverse can be computed after training the SchulzNN. For perturbed matrix derived from the original matrix, the network can be initialized with its pre-trained weights, enabling SchulzNN to achieve high-accuracy approximations of the inverse of perturbed matrix within few epochs—significantly reducing computational costs.

## 2 RELATED WORKS

**Learning to Optimize with Neural Networks**   The great success achieved by machine learning (ML) is backed up by the development and application of optimization techniques Tang & Yao (2024). In the field of Learning to Optimize (L2O), two major directions have emerged. The first direction integrates traditional optimization methods with neural networks. This approach combines the strengths of classical optimization techniques with the flexibility of machine learning models, allowing for more efficient optimization processes. By leveraging the power of neural networks, this hybrid method can automatically learn optimal strategies for solving specific types of optimization problems, thereby reducing the need for manual intervention and iterative processes. It has proven especially effective in domains where traditional optimization algorithms might struggle due to the complexity or size of the problem space. The second direction focuses on an end-to-end learning approach, where a neural network directly learns to predict the solution to an optimization problem from input data. In this fashion, the entire optimization process is treated as a learning task, bypassing traditional algorithmic steps entirely. The network is trained to output the solution in a manner similar to supervised learning. This approach has been particularly successful in combinatorial optimization problems, where the optimization task involves finding the best configuration from a large set of possibilities. By directly learning the solution mapping, this end-to-end approach has demonstrated impressive results in solving problems such as the traveling salesman problem and graph partitioning.

**Traditional Numerical Methods for Matrix Inverse.**   The general numerical method for finding the inverse of a matrix is Gauss-Jordan Elimination, which has a computational complexity of $O(N^3)$ with the matrix size $N$. Besides this method, there are also the LU decomposition method and the QR decomposition method, both of which have the same computational complexity as the Gauss-Jordan elimination method. When $N$ is large, the computational cost will be extremely high. The iterative methods are typified by Schulz iteration. The Schulz iteration algorithm first seeks a suitable initial matrix $A_0$, and then, through iterative calculations based on the second-order formula, it can rapidly approach the matrix $A_n$ that represents $A^{-1}$ Ben-Israel (1965). Based on the second-order form of Schulz iteration formula, formulas of the third-order form Li et al. (2011), the sixth-order form Krishnamurthy & Sen (1986), and the seventh-order form Soleymani (2012) have been proposed, which are able to have higher-order convergence as the order of the formula increases. This series of iterative algorithms is suitable for parallel computation and converges quickly, but finding the right $A_0$ is very difficult. Other methods focus on solving the inverse of specific matrices, such as symmetric positive definite matrices Finta (2020), integer matrices Haramoto & Matsumoto (2009), tridiagonal matrices and pentagonal matrices Hadj & Elouafi (2008). However, these methods are not universal.

**Neural Network Methods for Matrix Inverse.**   Most of the neural network methods focus on solving the inverse of time-varying matrices, which are represented by the zeroed neural networks and Zhang neural networks. As a special type of recurrent neural network, the zeroed neural network can be used to solve the Sylvester equation when the matrix is time-varying. And the task of solving the inverse of a time-varying matrix is a subtopic of solving the Sylvester equation Zhang et al.

(2002). With more and more people conducting research on the zeroed neural network and its related engineering applications, many variants of it have been successively proposed Xiao et al. (2019); Gao et al. (2024); Xiao et al. (2024). In 2012, another major representative neural network method, the Zhang neural network, was proposed. It is specialized in on-line inverse solving of time-varying matrices and can be applied to the motion control of robots Guo & Zhang (2012). Improved Zhang neural networks, such as NTZNN, have better noise resistance Xiang et al. (2018), and FTZNN can accelerate the convergence Xiao (2016), etc. Other methods focus on solving the Moore-Penrose inverse (M-P inverse) of complex matrices, etc. Song & Yam (1998); Xia et al. (2016). All these methods have great difficulty in solving the inverse of large-scale matrices.

## 3 METHODOLOGY

### 3.1 SCHULZ ITERATION

In 1933, Günther Schulz introduced the concept of iterative matrix inversion Schulz (1933). In 1965, Adi Ben-Israel refined Schulz's algorithm and established its convergence properties Ben-Israel (1965). While Schulz iteration extends to computing generalized inverses, this work focuses exclusively on computing inverses of full-rank matrices over the real field.

**Theorem 1 (Schulz Iteration)** *Let $A, A_0 \in R^{N \times N}$, and $A$ be invertible. $A_0$ satisfies the following conditions:*

$$A_0 = A^T B_0 = C_0 A^T \text{ and } \|AA_0 - I\| < 1, \|A_0 A - I\| < 1, \tag{1}$$

*where $B_0, C_0 \in R^{N \times N}$ and $B_0, C_0$ be invertible. $\| \cdot \|$ denotes the $L_2$-norm of the matrix. The Schulz iterative formula is:*

$$A_{n+1} = (2I - A_n A)A_n. \tag{2}$$

*Then a conclusion holds:*

$$\lim_{n \to \infty} \|A_n A - I\| = \lim_{n \to \infty} \|AA_n - I\| = 0. \tag{3}$$

Eq. (2) gives the formula of Schulz iteration, i.e., finding the $A_0$ that satisfies the conditions (1) and then iterate for a sufficient number of times based on Eq. (2), and an approximation of $A^{-1}$ with extremely high precision can finally be obtained. However, finding an $A_0$ that can successfully converge to $A^{-1}$ is a very challenging task, which significantly restricts the application scope of Schulz iteration.

### 3.2 SCHULZ NEURAL NETWORK

In order to expand the applicability of Schulz iteration, we propose the SchulzNN by imitating the structure of Schulz iteration. The structure of the SchulzNN is illustrated in Fig. 1.

Suppose that our task is to approximate the inverse matrix $A^{-1}$ of the matrix $A \in R^{N \times N}$. First, we need to randomly generate a certain number of $N$-dimensional vectors $b$ as the training set, and then feed them into the SchulzNN. The goal is to make the output vector $\hat{x}$ as close as possible to $A^{-1}b$. The SchulzNN consists of three layers. Each layer contains $N$ neurons. The weight matrix of the

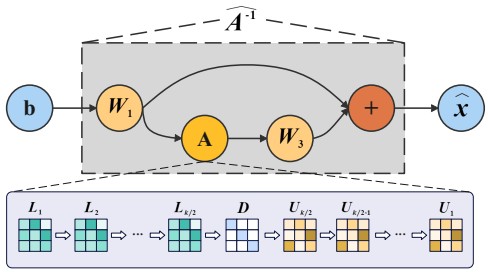

Figure 1: The structure of SchulzNN

$i$-th layer is denoted as $W_i(i = 1, 2, 3)$, and the corresponding output is $o_i$. Notably, there are no activation functions in the SchulzNN.

The weight matrices of the first and third layers are initialized to the same matrix $A_0$, while the second layer's weight matrix is initialized as $A$. During training, $W_2$ remains fixed and does not undergo updates via gradient backpropagation. We refer to the this layer parameterized by $W_2$ as the "$A$-layer". Accordingly, $W_2$ is denoted directly as $A$.

A key design feature of SchulzNN is its efficient handling of sparse or low-rank matrices $A$. By constructing the IDBF Tu et al. (2022) of $A$ and leveraging its structure for matrix-vector operations, the computational complexity of multiplying $A$ with a vector $v$ is reduced to $O(N \log N)$ instead of $O(N^2)$. This is achieved through the factorization:

$$A \approx L_1 L_2 \cdots L_{k/2} D U_{k/2} \cdots U_2 U_1. \tag{4}$$

The decomposition depth is $k = O(\log N)$, and all factors $L_i, U_i$ $(i = 1, 2 \cdots k/2)$ are sparse matrices containing $O(N)$ nonzero entries. Consequently, the original single layer of the neural network is expanded into distinct layers, each corresponding to one factor in the decomposition. Throughout the remainder of this paper, any reference to the matrix $A$ within the network architecture implies its IDBF representation.

The final output of SchulzNN is $\hat{x} = 2o_1 - o_3$. The loss function $\mathcal{L}$ of the SchulzNN is defined as the relative error between $A\hat{x}$ and $b$, i.e.,

$$\mathcal{L} = \frac{1}{m} \sum_{i=1}^{m} \frac{\|A\hat{x}_i - b_i\|}{\|b_i\|}, \tag{5}$$

where $m$ denotes the batch size and $\|\cdot\|$ represents the $L_2$-norm of the vector.

During the process of training, we first input the vector $b$. After it passes through the three fully connected layers, we obtain $o_1 = W_1 b$, $o_2 = AW_1 b$, $o_3 = W_3 AW_1 b$ respectively. The final output is

$$\hat{x} = 2o_1 - o_3 = 2W_1 b - W_3 AW_1 b. \tag{6}$$

We take the initialization values $W_1 = W_3 = A_0$ into the above equation and obtain:

$$\hat{x} = 2A_0 b - A_0 A A_0 b = (2I - A_0 A) A_0 b. \tag{7}$$

The right-hand side of Eq. (7) is very close to the form of Eq. (2), and it can be regarded as the NN form of a single Schulz iteration. The $A_0$ here corresponds to the $A_0$ used in the Schulz iteration. Our task is to make $(2I - W_3 A)W_1$ approach to $A^{-1}$ as closely as possible. Clearly, when $W_1, W_3 \to A^{-1}$, we have $(2I - W_3 A)W_1 \to A^{-1}$. In this way, a single Schulz iteration can be successfully simulated in the SchulzNN. Most importantly, after sufficient and effective training, we can get the approximation of $A^{-1}$:

$$\widehat{A^{-1}} = (2I - W_3 A)W_1. \tag{8}$$

One of the significant advantages of the SchulzNN is its inherently self-contained residual connection structure. It outputs a linear combination of the output $o_1$ of the first layer and the output $o_3$ of the third layer, and this output form is determined by Eq. (2). The residual connection structure of the SchulzNN is beneficial for the training of the deep NN.

Another notable advantage of the SchulzNN is the simplicity of constructing the dataset: we only need to randomly generate a certain number of vectors $b$, where $b \in R^N$. In practical applications, we just have information about $A$ and lack information about $A^{-1}$. This situation makes it impossible for us to generate a dataset containing $(b, x)$ data pairs due to the reason that according to the fitting objective of the SchulzNN, the output $\hat{x}$ should be close to $x = A^{-1}b$, which directly involves $A^{-1}$. Through a clever design, we transform the supervised task into an unsupervised task. Instead of making $\hat{x}$ approach $x$ as closely as possible, we make $A\hat{x}$ approach $b$, which is directly shown in Eq. (5). In terms of the results, these two tasks are almost equivalent. This is because when $\hat{x} \to A^{-1}b$, $A\hat{x} \to AA^{-1}b = b$, and for the latter case, we only need to use $b$ as the training set. This design scheme significantly reduces the difficulty of dataset generation and the amount of data, which makes the training of the SchulzNN more straightforward.

## 3.3 DEEP SCHULZ NEURAL NETWORK

The SchulzNN introduced in Subsection 3.2 is the fundamental SchulzNN structure, which corresponds to a single Schulz iteration. To enhance the fitting performance, we introduce the concept of a deep SchulzNN: $\text{SchulzNN}_k$. Here, $k$ represents the number of Schulz iterations corresponding to the SchulzNN model.

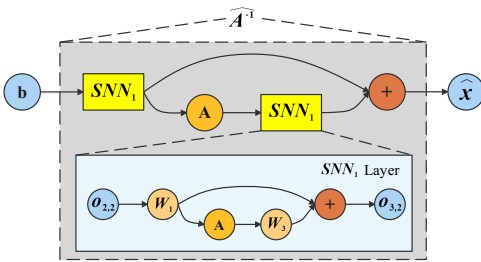

Figure 2: The structure of $\text{SchulzNN}_2$. $\text{SNN}_1$ denotes $\text{SchulzNN}_1$ for the sake of simplification.

Taking $\text{SchulzNN}_2$ as an example, it consists of two $\text{SchulzNN}_1$ layers and one $A$-layer in the middle, which is illustrated in Fig. 2. The structure of each $\text{SchulzNN}_1$ layer is exactly the same as that described in Subsection 3.2. The output of $\text{SchulzNN}_2$ is a linear combination of the outputs of the two $\text{SchulzNN}_1$ layers.

Let the weight of each Convolution Layer in the first $\text{SchulzNN}_1$ layer be denoted as $W_i^{(1)}$ and its output be $o_{i,1}^{(1)}(i = 1, 2, 3)$, and for the second $\text{SchulzNN}_1$ layer, let the weight be $W_i^{(2)}$ and the corresponding output be $o_{i,1}^{(2)}$. During the training process, we input the vector $b$. After passing through the first $\text{SchulzNN}_1$ layer, we obtain $o_{1,2} = 2o_{1,1}^{(1)} - o_{3,1}^{(1)}$. Then, it passes through the $A$-layer and becomes $o_{2,2}$, and after going through the second $\text{SchulzNN}_1$ layer, we get $o_{3,2} = 2o_{1,1}^{(2)} - o_{3,1}^{(2)}$. Finally, the output is $\hat{x} = 2o_{1,2} - o_{3,2}$. Assume that the first $\text{SchulzNN}_1$ layer is equivalent to $A_1^{(1)}$ and the second $\text{SchulzNN}_1$ layer is equivalent to $A_1^{(2)}$. When we expand the output, we have:

$$
\begin{aligned}
\hat{x} &= 2o_{1,2} - o_{3,2} \\
&= 2(2o_{1,1}^{(1)} - o_{3,1}^{(1)}) - (2o_{1,1}^{(2)} - o_{3,1}^{(2)}) \\
&= 2\left(2I - W_3^{(1)}A\right)W_1^{(1)}b - \left(2I - W_3^{(2)}A\right)W_1^{(2)}A\left(2I - W_3^{(1)}A\right)W_1^{(1)}b \\
&= (2I - A_1^{(2)}A)A_1^{(1)}b.
\end{aligned}
\tag{9}
$$

When $W_1^{(1)}, W_3^{(1)}, W_1^{(2)}, W_3^{(2)} \to A^{-1}$, we have $A_1^{(1)}, A_1^{(2)} \to A^{-1}$, which concludes $(2I - A_1^{(2)}A)A_1^{(1)} \to A^{-1}$, so that two Schulz iterations can be successfully simulated in the $\text{SchulzNN}_2$. And we finally get the approximation of $A^{-1}$:

$$
\widehat{A^{-1}} = (2I - A_1^{(2)}A)A_1^{(1)}.
\tag{10}
$$

With the structure of $\text{SchulzNN}_2$ obtained, we can recursively deduce the structure of $\text{SchulzNN}_k$ where $k \geq 2$, which is illustrated in Fig. 3.

The $\text{SchulzNN}_k$ model contains two $SchulzNN_{k-1}$ layers and one $A$-layer in the middle. In fact, $\text{SchulzNN}_k$ has $2^{k-1}$ $SchulzNN_1$ layers, and each $NN_1$ layer consists of three fully-connected layers. According to the order of the network structure, let the weights of the first and the last Convolution layers of the $i$-th $\text{SchulzNN}_1$ layer be $W_1^{(i)}$ and $W_3^{(i)}$ respectively. For a specific $p$ ($p = 1, 2, \cdots, k-1$), $\text{SchulzNN}_k$ contains $2^{k-p}$ $SchulzNN_p$ layers. For the $q$-th $SchulzNN_p$ layer in the network order, with $q = 1, 2, \cdots, 2^{k-p}$, there is an $A$-layer in the middle. We initialize all $W_1^{(i)}, W_3^{(i)}$ to $A_0$.

When $p < k - 1$, we define $m_q = 3 - 2 \times (q \mod 2)$ and $n_q = [\frac{q+1}{2}]$. The output of the $q$-th $SchulzNN_p$ layer is $o_{m_q,p+1}^{(n_q)} = 2o_{1,p}^{(q)} - o_{3,p}^{(q)}$. When $p = k - 1$, the output of the first $SchulzNN_{k-1}$ layer is $o_{1,k} = 2o_{1,k-1}^{(1)} - o_{3,k-1}^{(1)}$, and that of the second $SchulzNN_{k-1}$ layer is $o_{3,k} = 2o_{1,k-1}^{(2)} - o_{3,k-1}^{(2)}$. Finally, SchulzNN$_k$ outputs $\hat{x} = 2o_{1,k} - o_{3,k}$.

# 4 NUMERICAL EXPERIMENTS

## 4.1 SETTINGS FOR EXPERIMENTS

Our experimental task is as follows: Given a known matrix $A_{N \times N}$, use the SchulzNN model to approximate $A^{-1}$ by training. In this Section, $N = 1024$. We will randomly generate $S$ vectors of dimension $N$ as the training set. For different tasks, the size of the training set $S$ varies, and the batch size is set to be $0.0125S$. Different tasks also involve different choices of $A_0$, learning rate (lr) and learning rate decay. Common choices of $A_0$ are $\mathbf{0}$, $[\text{diag}(A)]^{-1}$ and $\text{diag}(A^{-1})$, where $\text{diag}(X)$ denotes a diagonal matrix whose diagonal elements are equal to the corresponding diagonal elements of $X$. The first two components are straightforward to compute, while the third can be efficiently computed using the method in Tu et al. (2022), which achieves $O(N)$ complexity for extracting the diagonal of the matrix inverse.

In order to accelerate the convergence efficiency, we employ the ADAM optimizer, which is widely utilized in deep learning, throughout the training process. The training is carried out on an NVIDIA RTX 4090 GPU.

After training, we need to assess the fitting performance of the SchulzNN on matrix inversion. In the inference phase, we assume that $A^{-1}$ is known (whereas $A^{-1}$ is unknown throughout the training). Suppose that our trained model is SchulzNN$_k$. Obviously, we can use the form like Eq. (8) and Eq. (10) to calculate the fitted $\widehat{A^{-1}}$, but for different $k$, the formula is changing. Instead, we use a special test set to calculate the $\widehat{A^{-1}}$ that can be simulated by $SchulzNN_k : \{e_i\}_{i=1}^N$, where

$$e_i = \{\overbrace{0, 0, 0, \cdots, 0}^{i-1}, 1, 0, \cdots, 0\}^T. \tag{11}$$

By inputting $e_i$ into the model SchulzNN$_k$, it returns the output $x_i$. The final expected output of SchulzNN$_k$ is $A^{-1}e_i$, which precisely corresponds to the $i$-th column of $A^{-1}$. Then, we perform the following operation to obtain $\widehat{A^{-1}}$:

$$\widehat{A^{-1}} = \text{concat}\,(x_1, x_2, \cdots, x_N)\,, \tag{12}$$

where concat$(\cdots)$ means column-wise concatenation.

The metric for the fitting effect is the relative error between $\widehat{A^{-1}}$ and $A^{-1}$, i.e.,

$$\epsilon_{inv} = \frac{\|\widehat{A^{-1}} - A^{-1}\|}{\|A^{-1}\|}. \tag{13}$$

In practice, $A^{-1}$ is unknown during the whole experiment, so we use another metric:

$$\epsilon_{sub} = \frac{\|A\widehat{A^{-1}} - I\|}{\|I\|}. \tag{14}$$

The training objective is to make $\epsilon_{inv}$ reach an order of magnitude of approximately $10^{-2}$ or $10^{-3}$, and make $\epsilon_{sub}$ reach an order of magnitude of $10^{-3}$.

## 4.2 NUMERICAL INSTANCES

In this section, we will apply SchulzNN to 4 different matrices to fit their inversion.

### 4.2.1 PERFORMANCE TEST

We start with a performance test for SchulzNN. The matrix for the performance test is a strict diagonal-dominance matrix, which has a very low condition number. That means the task is relatively simple. The goal of the task is to demonstrate the effectiveness of SchulzNN and the model utilized is the simplest SchulzNN$_1$ model. The information of the matrix is shown in Table 7.

It is worth noticing that the condition number is presented in the Table. In fact, after conducting extensive experiments, we found that the difficulty of the fitting task exhibits a relatively positive correlation with the matrix's condition number. Specifically, a larger condition number implies a more ill-conditioned matrix, making its inverse more challenging to fit. In this experiment, the condition number is 1.359, indicating that the task is relatively simple.

The size of the training set is $S = 1280$. In order to enable the model to converge more effectively, we have implemented a learning rate decay mechanism: the learning rate is reduced to half of its original value every $q$ epochs, which can be briefly denoted as $q/0.5$. The experiment consists of two parts: No.1 uses $\mathbf{0}$ as the initialization matrix and No.2 uses $\text{diag}(A)^{-1}$. For initial lr, No.1 is 8e-6 and No.2 is 1e-6. For learning rate decay, No.1 is 50/0.5 and No.2 is 20/0.5. For the number of training epochs, No.1 is 400 and No.2 is 100.

The Learning curves are presented in Fig. 4 and Fig. 5. Here, the horizontal axis denotes the number of training epochs, and the vertical axis represents the logarithm to base 10 (lg) of the training loss.

From Fig. 5, we can observe that the initial training loss in Experiment No.2 is already low. Starting with a better choice of $A_0$, the SchulzNN model can converge more quickly and effectively. In fact, when $\text{diag}(A)^{-1}$ is taken as $A_0$, the Schulz iteration can converge to an extremely high precision within just a few epochs. Therefore, this experiment serves merely to demonstrate the effectiveness of the SchulzNN, and the advantages of the SchulzNN will be presented in the subsequent sections. From another perspective, when matrix $\mathbf{0}$ is used as $A_0$, the Schulz iteration fails to operate, whereas the SchulzNN can achieve convergence, which partially illustrates the superiority of the SchulzNN.

| No. | Training loss | $\epsilon_{inv}$ | $\epsilon_{sub}$ |
|-----|---------------|------------------|------------------|
| 1   | 4.45e-3       | 4.74e-3          | 4.8e-3           |
| 2   | 4.59e-3       | 4.52e-3          | 4.66e-3          |

Table 1: The results of the experiment

According to the results of Table 1, it can be concluded that the SchulzNN model is effective. Through sufficient training, the inverse of the strictly diagonally dominant matrix $A$ can be approximated with a high precision of $10^{-3}$.

### 4.2.2 PERMUTATION MATRIX

From this subsection, we will focus on tasks that the Schulz iteration fails to solve (i.e., the iteration doesn't converge to $A^{-1}$ when taking the common $A_0$ as initialization), yet can be completed by the SchulzNN through training. The information of the matrices is presented in Table 7.

First, we will focus on a simple yet special matrix : the permutation matrix. As we can see in Table 7, it features a controllable parameter $k$.

The experiments consist of $k = 512$ and $k = 1024$, respectively. The initialization matrix is $\text{diag}(A^{-1})$ and the model SchulzNN$_1$ is used. The size of training set is 1280 and the number of training epochs is 500. The initial lr is 1e-3 and the lr decay is 50/0.5.

The learning curves are presented in Fig. 6 and Fig. 7. According to the results of Table 2, it can be observed that the SchulzNN model can converge to a high precision rapidly when dealing with the task of fitting the inverse of the permutation matrix, and it shows good adaptability to the value of $k$. Despite the simplicity of the task, it fully illustrates the advantages of the SchulzNN.

### 4.2.3 DISCRETE HELMHOLTZ MATRIX

In this subsection, we focus on another matrix, the Discrete Helmholtz matrix, and we attempt to fit it by using both the single-layer SchulzNN and the deep SchulzNNs.

| $k$ | **Training loss** | $\epsilon_{inv}$ | $\epsilon_{sub}$ |
|------|------|------|------|
| 512 | 5.05e-4 | 5.11e-4 | 5.11e-4 |
| 1024 | 5.14e-4 | 1.18e-3 | 1.18e-3 |

Table 2: The results of the experiment

According to Table 7, the condition number of the matrix is relatively high, which means the task is difficult to solve. This matrix represents the discrete Helmholtz matrix with wave number $k = 1$ and holds significant practical relevance in disciplines such as acoustic simulation, electromagnetism, and quantum mechanics, among others.

The initialization matrix is $\text{diag}(A^{-1})$ and the models used are $SchulzNN_k$ with $k = 1, 2, 3$. The size of training set is 5120 and the number of training epochs is 1500.

The learning curves are presented in Fig. 8 on the left. According to the results of Table 3, it can be observed that the single-layer SchulzNN and the $SchulzNN_2$ fail to adequately fit $A^{-1}$. With extremely low accuracy, it proves unsuitable for practical applications. Conversely, the $SchulzNN_3$, after training, demonstrates excellent fitting performance for $A^{-1}$, achieving an accuracy of $10^{-4}$. These findings highlight the superior fitting capabilities of some deep SchulzNNs.

| **Model** | **Initial lr** | **lr decay** | **Training loss** | $\epsilon_{inv}$ | $\epsilon_{sub}$ |
|------|------|------|------|------|------|
| $SchulzNN_1$ | 1.5 | 90/0.5 | 3.19e-2 | 0.111 | 3.78e-2 |
| $SchulzNN_2$ | 1e-3 | 25/0.93 | 1.67e-2 | 0.296 | 3.13e-2 |
| $SchulzNN_3$ | 1e-3 | 100/0.5 | 6.24e-5 | 1.16e-4 | 7.91e-5 |

Table 3: The settings and results of the experiment

### 4.2.4 PERTURBED IDENTITY MATRIX

In this subsection, the matrix that we focus on is the perturbed identity matrix, and its specific form is as follows:

$$I + uv^T. \tag{15}$$

Here, $u, v \in R^{N \times k}$, and $k \ll N$. We set $k = 5$. It is quite evident that $uv^T \in R^{N \times N}$ has a rank of $k$, and it plays the role of "perturbing" the identity matrix.

We can see in Table 7 that the condition number of this matrix is higher than that in Subsection 4.2.3.

The initialization matrix is $[\text{diag}(A)]^{-1}$ and the models used are $SchulzNN_k$ with $k = 1, 2, 3$. The size of training set is 5120 and the number of training epochs is 1800.

The learning curves are presented in Fig. 9 on the right. According to the results of Table 4, it can be observed that $SchulzNN_k$ with $k = 1, 2, 3$ can all effectively fit the perturbed identity matrix $I + uv^T$, achieving an accuracy approaching $10^{-3}$. Moreover, the deep SchulzNNs outperform the single-layer SchulzNN. This further validates the effectiveness of the SchulzNN model with deep architectures.

| **Model** | **Initial lr** | **lr decay** | **Training loss** | $\epsilon_{inv}$ | $\epsilon_{sub}$ |
|------|------|------|------|------|------|
| $SchulzNN_1$ | 6e-4 | 100/0.55 | 8.54e-3 | 6.04e-3 | 1.13e-2 |
| $SchulzNN_2$ | 2e-4 | 100/0.55 | 2.57e-3 | 2.68e-3 | 3.88e-3 |
| $SchulzNN_3$ | 4e-5 | 120/0.55 | 3.18e-3 | 3.61e-3 | 4.26e-3 |

Table 4: The settings and results of the experiment

### 4.3 PERTURBATION EXPERIMENT

Matrices are always perturbed due to various reasons. Under such circumstances, we expect to make use of the pre-trained SchulzNN models for fitting the inverse of matrices, i.e., through appropriate

fine-tuning, the models can converge to the inverse of the new matrices while maintaining a considerable level of accuracy. In this way, there is no need to initiate the training process from the very beginning, thus reducing a substantial amount of computational costs.

In this experiment, we will take the perturbation of the permutation matrix in Subsubsection 4.2.2, where $k = 512$, as an example to investigate the fitting effect after fine-tuning SchulzNN. The basic model is the SchulzNN$_1$ with the same settings in Subsubsection 4.2.2. Before the training process, we will input the new matrix $A$ into the $A$-layer of the base model and fix it.

There are 2 types of perturbation scenarios:

- Scenario 1: $A + \varepsilon I$;
- Scenario 2: $A + R$, $r_{ij} \in U(-\varepsilon, \varepsilon)$.

Before conducting the experiment, it's necessary to measure the magnitude of the perturbation. We denote the relative error between the inverse of matrix $A$ and the inverse of the perturbed matrix $\tilde{A}$ as $e_{A^{-1}}$. Below, we record the values of $e_{A^{-1}}$ for Scenario 1 and Scenario 2 under different values of $\varepsilon$, along with the condition numbers of the corresponding new matrices. Table 5 shows the measurement.

| | $\varepsilon$ | 0.6 | 0.9 | 0.95 |
|---|---|---|---|---|
| Scenario 1 | $e_{A^{-1}}$ | 0.593 | 1.5 | 2.196 |
| | Condition number | 4 | 19 | 39 |
| | $\varepsilon$ | 0.02 | 0.04 | 0.05 |
| Scenario 2 | $e_{A^{-1}}$ | 0.398 | 1.097 | 2.428 |
| | Condition number | 2.961 | 14 | 76.17 |

Table 5: The measurement of perturbation

| Scenario | No. | $\varepsilon$ | Initial lr | lr decay | Epochs | Training loss | $\epsilon_{inv}$ | $\epsilon_{sub}$ |
|---|---|---|---|---|---|---|---|---|
| | 1 | 0.6 | 1e-3 | 10/0.5 | 80 | 1.5e-3 | 1.01e-3 | 1.55e-3 |
| 1 | 2 | 0.9 | 1e-3 | 10/0.5 | 100 | 9.54e-4 | 3.69e-3 | 1.1e-3 |
| | 3 | 0.95 | 2e-3 | 20/0.5 | 100 | 2.89e-2 | 0.109 | 3.14e-2 |
| | 4 | 0.02 | 4e-4 | 10/0.5 | 50 | 3.83e-3 | 2.73e-3 | 3.92e-3 |
| 2 | 5 | 0.04 | 1e-3 | 10/0.5 | 80 | 1.44e-3 | 6.06e-4 | 1.5e-3 |
| | 6 | 0.05 | 2e-3 | 20/0.5 | 100 | 2.76e-2 | 0.184 | 3.1e-2 |

Table 6: The settings and results of the experiment

According to the results of Table 6, it can be observed that under the experimental conditions, within a certain perturbation range (Experiments No. 1, 2, 4, 5), the fine-tuned SchulzNN$_1$ model can converge effectively to the inverse of the perturbed matrix, achieving an accuracy of $10^{-3}$ or higher. A plausible explanation is that the pre-trained weights serve as a suitable initial approximation $A_0$ for the new problem, which facilitates quick convergence. Conversely, beyond this range (Experiments No.3, 6), fine-tuning fails to yield effective convergence. This is likely because the pre-trained weights deviate significantly from the true inverse.

## 5 CONCLUSIONS

In this paper, a neural network architecture SchulzNN is proposed, which is inspired by the Schulz iteration framework. Through specialized training tasks, the network learns to approximate matrix inverses, effectively modeling the mapping between matrices and their inverses. Leveraging the inherently recursive structure of the Schulz iteration, SchulzNN supports both single-layer and multilayer architectures, offering flexible solutions to the matrix inversion problem. Experimental results demonstrate that SchulzNN outperforms traditional Schulz iteration across diverse matrix classes, surpassing its performance limitations. In scenarios involving matrix perturbations, the pre-trained weights of SchulzNN can be adapted to initialize a new network, enabling efficient and accurate approximation of the perturbed matrix inverse with minimal retraining.

ETHICS STATEMENT

This work proposes SchulzNN to obtain matrix inversion. While all research carries potential societal implications, we have not identified any specific ethical concerns that warrant particular emphasis in this study.

REPRODUCIBILITY STATEMENT

All experimental results presented in this paper are reproducible. The implementation code for ParetoRouter, including scripts for reproducing all experiments, will be made publicly available on GitHub upon acceptance of the paper.

LLM USAGE STATEMENT

Large language models (LLMs) were used in a limited capacity during the preparation of this manuscript, specifically for: (1) polishing written text, and (2) reviewing sentence syntax. We explicitly declare that no experimental data, analysis, or methodological content was generated or modified using LLMs.

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

## A  APPENDIX

### A.1  DETAILS ABOUT SCHULZNN$_k$

Details about SchulzNN$_k$ include its structure illustrated in Fig. 3 and algorithms 1 and 2. It is worth noticing that the symbols in Algorithm 2 are not completely the same as the symbols in Subsection 3.3 for the sake of simplicity.

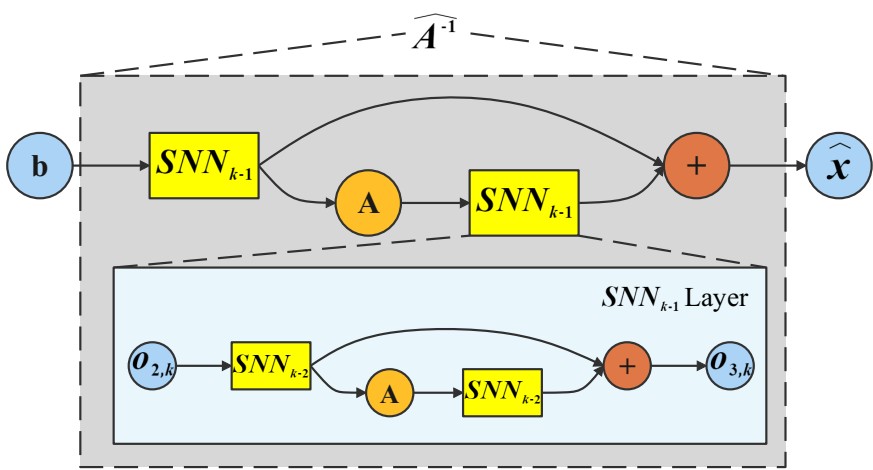

Figure 3: The structure of SchulzNN$_k$

---

**Algorithm 1:** The SchulzNN$_1$ Algorithm

---

**Input:** Vector $b$, Learning rate $\eta$
**Initialize** $W_1 = W_3 = A_0$
**while** not Converge **do**
    $o_1 = W_1 b$
    $o_2 = A o_1$
    $o_3 = W_3 o_2$
    $\hat{x} = 2 o_1 - o_3$
    **Update weights:** $W_1 \leftarrow W_1 - \eta \cdot \partial L / \partial W_1, \ W_3 \leftarrow W_3 - \eta \cdot \partial L / \partial W_3$
**end while**
**Output:** Vector $\hat{x}$

---

**Algorithm 2:** The SchulzNN$_k$ Algorithm

---

**Function:** Schulz_Recurrent_cal($o_{input}, p$)
**If** $p = 1$ **then**
    $o_{output} = SNN_1(o_{input})$
    **return** $o_{output}$
**else**
    $o_1 = $ Schulz_Recurrent_cal($o_{input}, p-1$)
    $o_2 = A o_1$
    $o_3 = $ Schulz_Recurrent_cal($o_2, p-1$)
    $o_{output} = 2 o_1 - o_3$
    **return** $o_{output}$
**end if Main Procedure:**
**Input:** Vector $b$, Learning rate $\eta$, num of Layers $k$
**Initialize** $W_1^{(q)} = W_3^{(q)} = A_0; q = 1, 2, \cdots, 2^{k-1}$
**while** not Converge **do**
    $\hat{x} = $ Schulz_Recurrent_cal($b, k$)
    **Update weights:** $W_1^{(q)} \leftarrow W_1^{(q)} - \eta \cdot \partial L / \partial W_1^{(q)}, \ W_3^{(q)} \leftarrow W_3^{(q)} - \eta \cdot \partial L / \partial W_3^{(q)}$
**end while**
**Output:** Vector $\hat{x}$

## A.2 Details about Experiments

Details about Experiments include the information of matrices in Section 4.2 shown in Table 7 and figures of learning curves in Section Section 4.2.

| Matrix | Generation method | Condition number |
|---|---|---|
| Strict

Diagonal-dominance Matrix | $A = L_1 \cdots L_5 D U_5 \cdots U_1$, where $U_i = L_i^T$; $L_1, \cdots, L_5, D$ are lower triangular matrices and diagonal matrix of matrices:
When $i \neq j$, $a_{ij} \sim U(-0.005, 0.005)$;
When $i = j$, $a_{ij} = \sum_{k=1}^{N} |a_{ik}| + \epsilon$, where $\epsilon \sim U(0, 0.01)$ | 1.359 |
| Permutation matrix | Randomly select $k$ rows of the identity matrix $I$
and swap their order, make it completely
different from the original order | 1 |
| Discrete Helmholtz matrix | $a_{ij} = \begin{cases} -1 & \text{if } i = j \\ 1 & \text{if } |i - j| = 1 \\ 0 & \text{otherwise} \end{cases}$ | 1694 |
| Perturbed identity matrix | $I + uv^T$, where $u, v \in M_{N \times k}$,
$u_{ij}, v_{ij} \sim U(-0.5, 0.5)$ | 4883 |

Table 7: The settings of the experimental matrices in Section 4.2

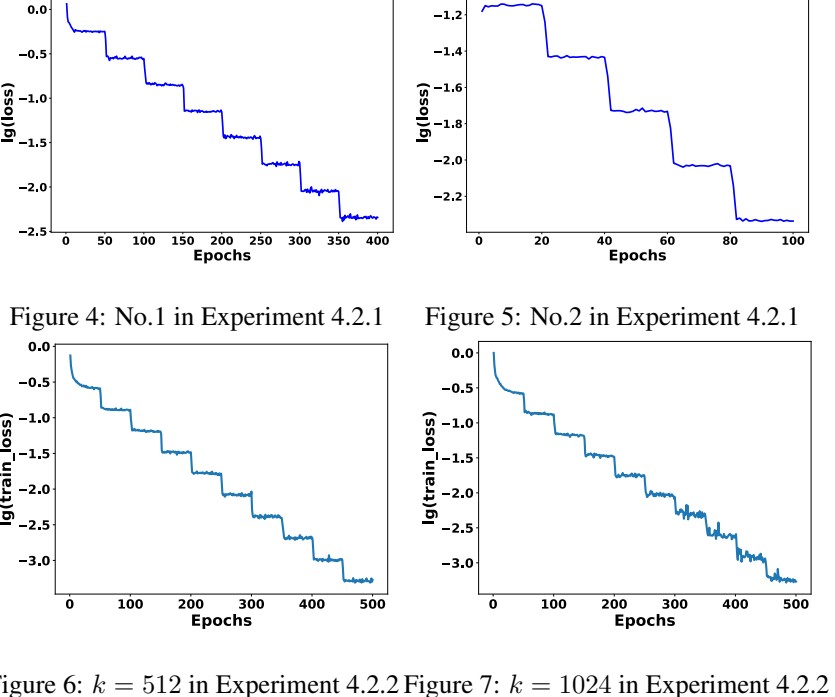

Figure 4: No.1 in Experiment 4.2.1    Figure 5: No.2 in Experiment 4.2.1

Figure 6: $k = 512$ in Experiment 4.2.2   Figure 7: $k = 1024$ in Experiment 4.2.2

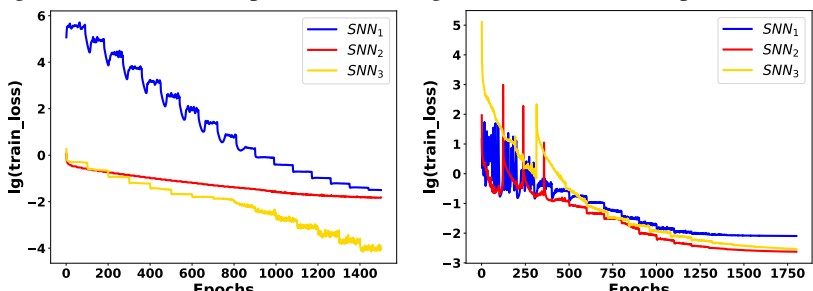

Figure 8: Experiment 4.2.3     Figure 9: Experiment 4.2.4