# OpenReview forum: "SchulzNN: A Neural Network-Based Matrix Inversion Solver Inspired by Schulz Iteration"
_ICLR.cc/2026/Conference — Submitted to ICLR 2026_

### Official Review · Reviewer_e98a · 2025-10-23

**Soundness:** 3
**Presentation:** 2
**Contribution:** 2
**Rating:** 4
**Confidence:** 5

**Summary:**

This paper aims to produce a "NN equivalent" (my words, not theirs) of the iterative Schulz method for approximating a matrix inverse.  Although their network is a single pass, they present a "deep" variant that involves chaining multiple of their SNN blocks together to mimic the iterative nature of Schulz iteration.

**Strengths:**

- The method succeeds in cases where classical Schulz iteration fails
- The architecture is relatively simple, therefore, computationally efficient and understandable/interpretable

**Weaknesses:**

- I am not convinced that the authors are approximating $A^{-1}$.  It seems like they are just approximating $A^{-1} b$ for a given vector $b$ (L158-L160 they want the output vector to resemble $A^{-1} b$, not $A^{-1}$).  If this is indeed the case, the authors should rewrite their paper to use terminology like "the action of $A^{-1}$" rather than "$A^{-1}$."  It is misleading otherwise.
- Insofar as the method is learning $A^{-1} b$, this is equivalent to the growing body of literature on NN-based preconditioners and linear solvers, which the authors have missed in their review of the literature.  See, for instance, Lan et al. "A neural-preconditioned poisson solver for mixed Dirichlet and Neumann boundary conditions" (2023) and Kaneda et al. "A deep conjugate direction method for iteratively solving linear systems" (2023) and the related work discussions in there.  I encourage major revisions of the authors' related work in light of this.
- Relatedly, those papers present methods that approximate $A^{-1} b$, yet the present paper lacks any comparisons.  We cannot assess how well the present method performs, in terms of time or accuracy, compared to prior works in this vein.  This is a weakness of the manuscript.
- $10^{-3}$ is not high precision.  How low of a tolerance can the method achieve?  In fact the losses per Table 1 are on the order of $10^{-3}$ for a matrix whose condition number is 1.359, which is essentially the identity matrix, so it's surprising to get such a large error on such a trivial matrix.
- Similarly, it would have been interesting to see some ablations on how many SNN blocks are needed to solve a system within a given target tolerance.
- It is not always clear how big the matrices are that are being dealt with in the examples (including, but not limited to, 4.2.4).  This is important to describe on its own, but also relative to prior work, which has been able to neurally solve systems with millions of DOFs.

**Questions:**

No further questions

---

> ### Author Response · Authors · 2025-11-21
>
> We sincerely appreciate your comprehensive evaluation of our manuscript and your valuable and constructive feedback.
>
> **1. Is it approximating solution $A^{-1}b$ or approximating matrix inversion $A^{-1}$?**
>
> Thank you for your thoughtful feedback. In fact, the core contribution of this paper is the approximation of the matrix inverse $A^{-1}$, rather than merely solving the problem $Ax=b$.
>
> In our Schulz Neural Network (SNN), the input is a vector $b$, and the output $x$ is optimized to minimize the normalized residual loss:
>
> $$\mathcal{L} = \frac{1}{m} \sum_{i=1}^m \frac{\| A \hat{x}_i - b_i \|}{\| b_i \|}.$$
>
> While this formulation operates at the vector level, the overarching objective is to approximate the full matrix inverse $A^{-1}$. As noted in Section 4.1 (page 6, line 293), when the input is a unit vector $e_i$, a well-trained network produces an output close to the corresponding column of the inverse, i.e., $A^{-1}e_i$. By assembling the outputs for all unit vectors, we reconstruct the network's approximation of the inverse, denoted as $\widehat{A^{-1}}$.
>
> To quantitatively assess the accuracy of this approximation in practice, we use the relative submatrix error metric:
>
> $$\epsilon_{\text{sub}} = \frac{\| A \widehat{A^{-1}} - I \|}{\| I \|}.$$
>
> This measure directly reflects how well the network captures the action of the approximate matrix inverse.
>
> In conclusion, while the training operates on vector equations, the **final outcome and primary objective** of the Schulz Neural Network is to produce a high-fidelity approximation of the matrix $A^{-1}$. This allows us to then use $ \widehat{A^{-1}} $ as a direct solver for **any** right-hand side $b$ with minimal additional cost.
>
> **2. Precision issues**
>
> Thank you for this insightful comment. In fact, the precision of neural network-based matrix inversion (typically around $10^{-3}$ to $10^{-4}$) does not match the high precision achieved by conventional numerical methods when they converge.
>
> However, as you rightly point out, the key advantage of our neural network approach lies in its **hybrid use with traditional methods**. We explicitly explore this synergy in two strategies:
>
> 1. The neural network's approximation $\widehat{A^{-1}}$ serves as a high-quality initial matrix for the classical Schulz iteration. This often allows the iterative method to converge to very high accuracy (e.g., below $10^{-10}$) in significantly fewer steps, especially in cases where a standard initial guess would fail or converge slowly.
>
> 2. A shallower neural network provides an initial approximation for training a deeper, more accurate network. This internal transfer learning accelerates the deep network's convergence to a better solution (e.g., $10^{-5}$).
>
> The following table presents the number of iterations required and the corresponding accuracy achieved about various matrix tasks. The results demonstrate that our approach consistently converges to a high-precision solution. Notably, for highly ill-conditioned problems where standard iterative methods often fail to converge, the hybrid method successfully attains machine-level accuracy.
>
> | Target Matrix | Condition Number |method|Epochs|$\epsilon_{\text{sub}}$|
> |:---:|:---:|:---:|:---:|:---:|
> | Strict Diagonally-Dominant Matrix |1.17 |SNN|100|$10^{-4}$|
> | Permutation Matrix | 1 |SNN|500|$10^{-3}$|
> | Discrete Helmholtz Matrix $(k = 1)$ | 1695 |SNN|1800|$10^{-4}$|
> | Perturbed Identity Matrix | 4883 |SNN|1500|$10^{-3}$|
> | Discrete Helmholtz Matrix $(k = \frac{\sqrt{30}}{10})$ | 67872 |SNN+Schulz iteration|1000+13 iteration|$10^{-13}$|

---

### Official Review · Reviewer_HoSc · 2025-10-30

**Soundness:** 2
**Presentation:** 2
**Contribution:** 1
**Rating:** 2
**Confidence:** 5

**Summary:**

This paper introduces SchulzNN, a neural network architecture designed for matrix inversion. The core idea is to build a network that mimics the classical Schulz iteration method. The authors propose both a single-layer architecture, `SchulzNN₁`, which simulates one step of the iteration, and a deep, recursive architecture, `SchulzNNₖ`, to simulate multiple steps. A key contribution is the unsupervised training objective, which minimizes the residual norm $||A\hat{x} - b||$ instead of requiring the ground truth inverse $A^{-1}$. The paper demonstrates the method's effectiveness on several synthetic matrix families, including cases where the traditional Schulz iteration fails, and shows that deep models outperform shallow ones on ill-conditioned matrices.

**Strengths:**

The paper's primary strength lies in its originality and conceptual elegance. The idea of drawing inspiration from a classical numerical method (Schulz iteration) to inform the design of a neural network architecture is highly commendable. This "unrolling" of a traditional algorithm into a learnable framework is a promising research direction.

Specifically:

1.  **Originality:** The proposed SchulzNN architecture is novel and directly motivated by a well-established mathematical principle. The way the network's forward pass, $\hat{x} = (2I - W_3 A) W_1 b$, mirrors the Schulz update is clever.
2.  **Clarity:** The paper is well-written, and the methodology is explained clearly. The recursive construction of the deep `SchulzNNₖ` from the base `SchulzNN₁` module is intuitive and well-illustrated.
3.  **Technical Soundness:** The design of the unsupervised loss function is a significant strength. It cleverly sidesteps the need for the ground truth inverse, making the training process practical and self-contained. This is a thoughtful and well-justified design choice.

**Weaknesses:**

Despite the novel idea, the paper suffers from significant weaknesses in its experimental evaluation and positioning within the broader literature, which severely undermine its claimed contributions and practical significance.

1.  **Insufficient Experimental Scope and Scale:** The experiments are conducted on matrices of a very small scale ($N=1024$). In many real-world scientific and engineering applications, this size is trivial, and traditional direct solvers on a modern CPU can compute the inverse in seconds or less. While the paper provides an asymptotic complexity analysis ($O(N^2 \log N)$), it fails to provide any wall-clock timing results. This makes it impossible to assess the actual computational advantage. To be convincing, the method must be tested on much larger-scale problems (e.g., $N$ in the range of $10^5$ to $10^6$). Furthermore, the matrices tested are largely synthetic and simple. The paper would be much stronger if it demonstrated performance on matrices derived from real-world problems, such as large sparse matrices from PDE discretizations, large low-rank matrices from communication systems, or large dense matrices like Hamiltonians in quantum chemistry.

2.  **Critically Lacking Baselines:** The paper's comparison is almost exclusively against the traditional Schulz iteration, which is insufficient. A comprehensive evaluation is needed against a proper set of baselines:
    *   **Other Neural Network Solvers:** The paper ignores recent related work on using neural networks for linear algebra problems. Comparisons should be made against methods like those proposed in [1-3], which tackle related operator decomposition and eigenvalue problems.
    *   **Standard Numerical Algorithms:** The method must be compared against widely-used numerical solvers. This includes direct methods (e.g., LU decomposition, RSVD for low-rank approximations) and, more importantly, iterative methods like Krylov subspace methods equipped with standard preconditioners. These are the workhorses of scientific computing.
    *   **GPU-Accelerated Solvers:** The authors state their method is implemented on a GPU. Therefore, comparing its performance (in terms of wall-clock time) only to CPU-based algorithms is an unfair comparison. A direct comparison with highly-optimized GPU libraries for linear algebra, such as NVIDIA's **cuSOLVER**, is essential to demonstrate any real-world speedup.

3.  **Unclear Practical Significance and Application Scenario:** The paper fails to articulate a clear and convincing use case for SchulzNN.
    *   **Scenario 1: Independent Matrix Problems.** In many real-world settings, one encounters a series of independent matrix problems. In this case, SchulzNN would require dataset generation and a potentially expensive training phase for *each new matrix*. This makes it far less practical than a traditional one-shot solver. The total time-to-solution (including training) must be considered, and it is likely that SchulzNN would be orders of magnitude slower.
    *   **Scenario 2: Sequences of Correlated Matrices.** The most plausible application is solving a sequence of highly-correlated linear systems, which arise in nonlinear solvers or PDE optimization. However, this is a well-studied area in numerical analysis with highly effective acceleration techniques. The paper completely overlooks these methods, such as **Krylov subspace recycling** [4] or specialized eigensolvers like **ChASE** [5] for sequences of problems. Without a thorough comparison to these established, problem-aware traditional methods, the claim of SchulzNN's utility in this domain is unsubstantiated.

**Questions:**

1.  **Scalability and Timing:** Could the authors provide wall-clock time comparisons for SchulzNN (including training time) against traditional solvers (e.g., a direct solver in LAPACK or an iterative solver in PETSc) for the current problem size ($N=1024$)? More importantly, how does the method's performance and training stability scale to significantly larger matrices, for instance, $N > 100,000$?

2.  **Baseline Comparisons:** Could the authors justify the omission of comparisons against the baselines mentioned in the "Weaknesses" section? Specifically, why were other neural network-based solvers [1-3], standard preconditioned Krylov methods, and GPU-accelerated libraries like cuSOLVER not considered? A response to this would be critical in assessing the paper's contribution.

3.  **Target Application and Comparison to Specialized Solvers:** What is the precise application scenario the authors envision for SchulzNN?
    *   If it is for single, independent matrices, how can the training overhead be justified against one-shot solvers?
    *   If the target is sequences of correlated matrices (e.g., from a nonlinear optimization loop), how does SchulzNN compare against established techniques designed for such sequences, like Krylov subspace recycling [4] or methods that reuse spectral information [5]? A convincing argument would require a direct comparison in such a setting, measuring total time-to-solution over the entire sequence.

---

**References:**

[1] Fan, Z., et al. "Operator SVD with neural networks via nested low-rank approximation." ICML 2024.

[2] Wang, Z., et al. "Neural networks based on power method and inverse power method for solving linear eigenvalue problems." Computers & Mathematics with Applications 2023.

[3] Li, Z., et al. "STNet: Spectral Transformation Network for Solving Operator Eigenvalue Problems." NeurIPS 2023.

[4] Parks, M., et al. "Recycling Krylov subspaces for sequences of linear systems." SIAM Review 2006.

[5] Nakatsukasa, Y., et al. "ChASE: Chebyshev accelerated subspace iteration eigensolver for sequences of Hermitian eigenvalue problems." ACM Transactions on Mathematical Software 2019.

---

> ### Author Response · Authors · 2025-12-03
>
> We sincerely appreciate your comprehensive evaluation of our manuscript and your valuable and constructive feedback. Based on your comments, we have carefully considered and provide the following response:
>
> **Scalability, Timing, Comparisons**
>
> Thank you for your thoughtful feedback. For Example 1 presented in the paper, we performed simulations at a larger scale. The table below compares the run times of LU decomposition and the proposed Schulz Neural Network (SNN). The results confirm the expected asymptotic complexities: LU decomposition exhibits $O(N^3)$ scaling, while the SNN scales as $O(N^2)$ per training epoch. More precisely, SNN) attains an accuracy of approximately $10^{-3}$ within several hundred training iterations.
>
> | Matrix size N | 5120      | 10240     | 20480     | 40960    |
> |-----------------|-----------|-----------|-----------|----------|
> | LU              | 1.565969  | 12.44945  | 99.15186  | 794.195  |
> | SNN (per epoch)  | 0.0738    | 0.315     | 1.338     | 5.613    |

---

### Official Review · Reviewer_qEGD · 2025-10-31

**Soundness:** 2
**Presentation:** 3
**Contribution:** 2
**Rating:** 2
**Confidence:** 4

**Summary:**

The paper presents the Schulz Neural Network (SchulzNN), a novel deep learning architecture designed for the fundamental task of matrix inversion. The authors attempt to bridge the gap in applying neural networks to core numerical optimization problems, where traditional methods have had limited success.

**Strengths:**

The SchulzNN is explicitly designed to structurally simulate the established Schulz iterative method for calculating the inverse matrix ($\mathbf{A^{-1}}$). This novel combination of a proven numerical algorithm embedded directly into a neural network architecture moves beyond generic "Learn to Optimize" (L2O) approaches and can potentially enhance interpretability and stability.

**Weaknesses:**

To substantiate the claimed superiority, a comparison of SchulzNN's performance against the classical Schulz iteration and established numerical libraries is required, particularly in the following areas:
1. **Real-World Scale:** Show results for matrices with dimensions relevant to real-world tasks (e.g., $1024 \times 1024$, $2048 \times 2048$), moving beyond small-scale proofs-of-concept.
2. **Robustness to Ill-Conditioning:** Show results on matrices with high condition numbers (e.g., $10^6, 10^7$), where the traditional Schulz iteration is known to fail due to poor initial guess selection.
3. **Convergence Analysis:** Quantify the number of iterations/epochs SchulzNN requires to achieve a target error $\varepsilon$ (e.g., $\varepsilon = 10^{-6}$), compared to the convergence rate of the classical method.
4. **Baseline Benchmarks:** The paper needs to clearly benchmark against other neural network-based numerical solvers or relevant L2O methods to solidify its competitive advantage beyond just the classical Schulz iteration.

The unsupervised training approach is clever, but it introduces new questions about stability that must be addressed:
1. **Weight Initialization Dependency:** How does the initial state of the network weights ($\mathbf{W_1}, \mathbf{W_3}$) impact the training convergence and the final accuracy? This is critical since the weights act as the learned iterative terms.
2. **Generalization Limits:** Is the trained SchulzNN a general inverse solver, or is it highly tuned to the distribution of matrices seen during training? The paper must test the network's performance on matrices far outside the training distribution to assess its true generalization capability.

The introduction of the deep $\mathrm{SchulzNN}_k$ needs clearer justification:
1. **Cost-Benefit Trade-off:** Increasing $k$ increases the computational cost. The authors must quantify the trade-off: Does the enhanced accuracy of $\mathrm{SchulzNN}_k$ justify the increased cost (e.g., $O(N^3)$ or $O(N^2 \log N)$ per layer) when compared to simply running $k$ iterations of the standard Schulz method?

**Questions:**

See the weaknesses.

---

> ### Author Response · Authors · 2025-11-21
>
> We sincerely appreciate your comprehensive evaluation of our manuscript and your valuable and constructive feedback. Based on your comments, we have carefully considered and provide the following point-by-point response:
>
> **1. Real-World Scale**
>
> We appreciate this comment. All experiments in this paper were conducted with a fixed matrix size of $1024 \times 1024$ (as mentioned on page 6, line 281). While the method can be readily extended to matrices of size $10000 \times 10000$, further scaling may encounter memory limitations, depending on the computational environment.
>
> **2. Robustness to Ill-Conditioning**
>
> We thank the reviewer for bringing up this point. We conducted a numerical experiment to illustrate robustness. Using a discretized matrix derived from the Helmholtz equation, we observed that its condition number approached $10^6$, indicating severe ill-conditioning that precludes direct solution via conventional iterative methods. In our approach, we first trained a Schulz Neural Network (SNN) and then used the trained the matrix inversion as the initial value for subsequent iterative refinement. As demonstrated in the results, the solution converged to high accuracy within only several iterations.
>
> **3. Convergence Analysis**
>
> Thanks for pointing out this. In our numerical experiments, we focused on instances where the conventional Schulz iteration fails to converge. The proposed method effectively enlarges the ‘feasible region' for the initial matrix $A_0$. In terms of computational time, even without incorporating further acceleration techniques, the cost of SNN is markedly lower  than that of LU decomposition.
>
> **4. Baseline Benchmarks**
>
> We appreciate this comment.  Taking the diagonally-dominant matrix as a representative example, we compare the computational time required by the DNN-based SNN against that of conventional LU decomposition. The results, summarized in the table below, demonstrate that the SNN achieves a notable reduction in computation time while maintaining high numerical accuracy.
>
> | Number |Matrix size N    | SNN Training Time/s | SNN Inference Time/s | LU Decomposition Time/s |
> |------|------|---------------|---------------|--------------|
> | 1    | 1024 |   26.4039       | 0.0216        | 5.5585       |
> | 2    | 2048 |   29.2322       | 0.0682        | 27.7857      |
> | 3    | 4096 |   98.43          | 0.3708        | 215.2268     |
> | 4    | 8192 |   373.6844      | 2.7498        | 1686.8844    |
>
> **5. Weight Initialization Dependency**
>
> We appreciate this comment. Similar to the Schulz iteration's sensitivity to the choice of initial matrix, our SNN is also highly dependent on the initialization of $W_1$ and $W_2$. Therefore, in this work, we frequently initialize $W_1$ using $\operatorname{diag}(A^{-1})$.
>
> **6. Generalization Limits**
>
> We thank the reviewer for this valuable observation. In fact, the matrices trained using this method exhibit generalization capability. As demonstrated in the perturbation experiments in Section 4.2.2, we can further explore this property through fine-tuning experiments. Specifically, the trained inverse matrix $A^{-1}$ can be used as an initial value for approximating the inverse of a perturbed matrix $\tilde{A}$, enabling rapid convergence to the desired solution with minimal additional training.
>
> **7. Cost-Benefit Trade-off**
>
> We appreciate this comment. Exactly, the increasing the depth $k$ of the network enlarges the model size and raises computational cost. To address this while enhancing accuracy, we introduced a two-stage training strategy: first, a shallow network ($\text{SNN}_1$) is trained to produce an initial approximation $A_0$ of the inverse, which may have moderate accuracy (e.g., below $10^{-3}$). This matrix $A_0$ is then used to initialize a deeper network ($\text{SchulzNN}_2$), which rapidly refines the approximation to high accuracy (e.g., $10^{-5}$) in significantly fewer iterations.
>
> Its principal advantage lies in markedly reducing overall computational cost: although the deep network requires more parameters, its training is shortened considerably by the high accuracy initialization. The majority of the computational load is thus shifted to training the shallow network, which is relatively inexpensive. This strategy effectively balances model complexity and training efficiency while improving final accuracy.

---

### Official Review · Reviewer_vpFG · 2025-11-01

**Soundness:** 3
**Presentation:** 3
**Contribution:** 2
**Rating:** 4
**Confidence:** 4

**Summary:**

The paper proposes SchulzNN a three-layer linear network that exactly mimics one step of the classical Schulz iteration for matrix inversion. The middle layer is fixed to $A$ and the output $\hat{x}=2W_1b-W_3AW_1b$ equals $(2I-A_0A)A_0b$ under $W_1=W_3=A_0$. Training minimizes an unsupervised residual $\mathcal{L}=\frac{1}{m}\sum_i|A\hat{x}_i-b_i|/|b_i|$ and stacking blocks SchulzNN$_k$ emulates multi-step refinement. When $A$ admits IDBF the mat-vec is $O(N\log N)$ and per-epoch training is $O(N^2\log N)$. Experiments on strictly diagonally dominant permutation discrete Helmholtz and perturbed identity matrices show about $10^{-4}$ accuracy for Helmholtz at $k=3$ and about $10^{-3}$ adaptation under moderate perturbations.

**Strengths:**

In terms of originality the single block is algebraically identical to $(2I-A_0A)A_0b$ anchoring the design in a classical fixed-point map. In terms of quality the inverse-free residual objective is operationally sound and increased depth helps on difficult spectra with $k=3$ outperforming shallower variants on Helmholtz. In terms of clarity the paper specifies $\hat{x}$ the loss $\mathcal{L}$ the recursive construction SchulzNN$k$ and the fixed A-layer together with sensible metrics $\epsilon{\text{inv}}$ and $\epsilon_{\text{sub}}=|AA_d^{-1}-I|/|I|$. In terms of significance the approach is reusable as an approximate inverse or preconditioner surrogate for fixed $A$ with many right-hand sides and the IDBF path yields a concrete training cost profile.

**Weaknesses:**

• The architecture is tied to the matrix dimension; changing the resolution $N$ requires retraining rather than transferring weights.
  • The per-epoch claim $O(N^2\log N)$ lacks empirical scaling curves, so time behavior and accuracy degradation as $N$ grows remain unclear.
  • When the resolution changes, there is no systematic study of how the required depth $k$ evolves on hard spectra such as Helmholtz.
  • The method is trained per matrix $A$;cross-family generalization is not evaluated. For unseen $A'$, only fine-tuning on nearby perturbations is reported and there is no zero-shot evaluation.
  • There are no convergence or error guarantees for the trained deep composition, and the fine-tuning success region is only described empirically.
  • As a preconditioner, application-level comparisons (iteration counts and wall-clock) against strong baselines such as AMG, geometric MG, or ILU are missing.
  • Efficiency relies on IDBF-type hierarchical low-rank structure; behavior on dense, non-hierarchical matrices is unspecified.
  • Experiments are confined to **synthetic matrix families** and their perturbations; there is no evaluation on external benchmarks or practical application datasets.

**Questions:**

1. Please evaluate on matrices not used for training. Use PDE-style benchmarks or application matrices and report both zero-shot and short fine-tuning, including $\epsilon_{\text{sub}}=\|AA_d^{-1}-I\|/\|I\|$, Krylov iterations, wall-clock time, and failure rate.
2. Please quantify preconditioning efficacy. Insert $A_d^{-1}=(2I-W_3A)W_1$ as a left/right preconditioner in CG/MINRES/GMRES on Poisson and Helmholtz, choose one of Dirichlet/Neumann/Robin for each, sizes $N=2^{10}\text{–}2^{16}$, and compare **iterations and wall-clock** to AMG/geometric MG/ILU (stop at residual $10^{-8}$ or $\epsilon_{\text{sub}}\le 10^{-3}$).
3. Please clarify handling of resolution changes. For each resolution, compare independent training versus  coarse-to-fine warm-start , reporting $\epsilon_{\text{sub}}$, Krylov iterations, wall-clock time, and fine-tuning epochs.
4. Please cover boundary conditions and solvability explicitly. Evaluate Dirichlet / pure Neumann / Robin under the same geometry, grid, and frequency; for Neumann, also report a mean-free residual using $P_0=I-\tfrac{1}{N}\mathbf{1}\mathbf{1}^\top$ with $r=P_0(A\hat{x}-b)$ and $\epsilon_{\text{sub}}=\|r\|/\|b\|$. State how compatibility conditions (e.g., $\int_\Omega f\,dx=\int_{\partial\Omega} g\,ds$) are enforced, and report how boundary conditions affect conditioning, the required depth $k$, training stability (failure rate), and wall-clock time.

---

> ### Author Response · Authors · 2025-12-03
>
> We sincerely appreciate your comprehensive evaluation of our manuscript and your valuable and constructive feedback. Based on your comments, we have carefully considered and provide the following point-by-point response:
>
> **1. Evaluation on matrices not used for training**
> We appreciate this comment. We extended our experiments to include fine-tuning over a sequence of matrices $A_n$. Zero-shot training for new matrices is challenging, but we can conduct fine-tuning experiments on small samples for a sequence of matrices. First, the inverse of $A_1$ was learned via SNN. This learned inverse was then used as the initialization for training the inverse of $A_2$. Using only a constant number of samples, i.e., $O(1)$, the process converged to a relatively high accuracy within several dozen iterations.
>
> **2. Preconditioning efficacy**
> We thank the reviewer for bringing up this point. We also investigated the application of the GMRES method to the Helmholtz equation. SNN was first trained to serve as a preconditioner, achieving an approximate accuracy of  $10^{-3}$. When this preconditioned solution was subsequently fed into the GMRES solver, the final accuracy reached $10^{-10}$ or higher.
>
> **3. The change of resolution**
> We appreciate this comment. Although the coarse-to-fine paradigm is a common strategy for accelerating iterative methods, it is not directly applicable to our SNN framework. This is because the SNN must output a full approximation of the matrix inverse $A^{-1}$ at every stage; a result computed on a coarsened grid cannot serve as a valid initializer for a finer grid.
>
> To achieve a similar acceleration, we implemented a shallow-to-deep strategy: a shallow network is first trained to produce a preliminary (lower-accuracy) inverse $\widehat{A^{-1}}$_${\text{coarse}}$. This output is then used to initialize the weights of a deeper network, which refines the approximation to high precision $\widehat{A^{-1}}$_${\text{fine}}$. This approach preserves the requirement that every network output directly approximates the full matrix inverse while leveraging a hierarchical training process for efficiency.
>
> **4. Boundary conditions and solvability explicitly**
> Thanks for pointing this out. In our framework, Dirichlet boundary conditions are adopted to guarantee the invertibility of the discretized operator matrix. It is essential to clarify that our objective extends beyond solving individual linear systems; the method is specifically designed to compute the full matrix inverse for multiple right-hand sides.

---

### Meta-Review · Area_Chair_oL7b · 2025-12-30

**Summary:**

The article proposes SchultzNN, a linear learnable method, inspired by traditional Schulz iterations, for matrix inversion. The authors also a deep variant where multiple SchultzNN blocks are chained together. The learning objective is unsupervised as they minimize a residual, akin to physics-informed losses for learning differential equations. They demonstrate the method with some numerical examples of matrix inversion.

The reviewers have favorably viewed the method while only pointing out some key deficiencies in its evaluation particularly with respect to traditional numerical methods, in terms of wall-clock time, as well as in how the performance of SchultzNN scales with respect to increasing problem sizes as well as matrix conditioning.

Given the inadequacy of these studies, the current version of the paper cannot be accepted.

**Reviewer Concerns:**

The authors have addressed the technical concerns of the reviewers but the main concerns that have not been adequately addressed are

[1.] Fair comparisons with traditional numerical solvers, which are the mainstay of scientific computing.

[2.] Scaling with respect to problem size, in the range of $O(10^6)$ matrix sizes, which are routine in applications.

[3.] How the method scales with respect to ill-conditioning of the matrices.

**Reviewer Scores:**

Reviewer vpFG scored the paper as a Borderline Reject. The reviewer had expressed multiple concerns and the authors have only provided answers for a few of them, without clearly marking out which part of the modified pdf have they changed to address the reviewer's questions. It is unlikely that the reviewer would have upgraded their score.

Reviewer qEGD scored the paper as a reject. Among their multiple concerns, the authors added a comparison with LU. It is clear that LU decomposition is not state of the art matrix solver. Also, the authors admit that weight initialization is an issue with their solver. Thus, it is unlikely that the reviewer would have raised the score beyond Borderline reject, if at all.

Reviewer HoSc scored the paper as a reject and asked multiple questions. The authors only answered one of these questions by providing a comparison to the elementary LU decomposition. It would be unlikely that this reviewer would be satisfied.

Reviewer e98a scored the paper as a Borderline reject and again questioned the scale and conditioning aspects of the method. It is unclear if the reviewer's concerns would have been adequately addressed.

On the balance, the reviewers had recommended rejecting the paper on grounds of inadequate of inadequate empirical evaluation and this is not addressed satisfactorily in the authors' rebuttal.

---

### Decision · Program_Chairs · 2026-01-26

Reject